# Molecular Mechanisms and Bioavailability of Polyphenols in Prostate Cancer

**DOI:** 10.3390/ijms20051062

**Published:** 2019-03-01

**Authors:** Teodora Costea, Péter Nagy, Constanța Ganea, János Szöllősi, Maria-Magdalena Mocanu

**Affiliations:** 1Department of Pharmacognosy, Phytochemistry and Phytotherapy, “Carol Davila” University of Medicine and Pharmacy, 020021 Bucharest, Romania; teodora.costea@umfcd.ro; 2Department of Biophysics and Cell Biology, Faculty of Medicine, University of Debrecen, 4032 Debrecen, Hungary; nagyp@med.unideb.hu; 3Department of Biophysics, “Carol Davila” University of Medicine and Pharmacy, 020021 Bucharest, Romania; constanta.ganea@gmail.com; 4MTA-DE Cell Biology and Signaling Research Group, Faculty of Medicine, University of Debrecen, 4032 Debrecen, Hungary

**Keywords:** dietary polyphenols, bioavailability, molecular mechanisms, prostate cancer

## Abstract

Prostate cancer is the one of the most frequently diagnosed cancers among men over the age of 50. Several lines of evidence support the observation that polyphenols have preventive and therapeutic effects in prostate cancer. Moreover, prostate cancer is ideal for chemoprevention due to its long latency. We propose here an equilibrated lifestyle with a diet rich in polyphenols as prophylactic attempts to slow down the progression of localized prostate cancer or prevent the occurrence of the disease. In this review, we will first summarize the molecular mechanisms of polyphenols in prostate cancer with a focus on the antioxidant and pro-oxidant effects, androgen receptors (AR), key molecules involved in AR signaling and their transactivation pathways, cell cycle, apoptosis, angiogenesis, metastasis, genetic aspects, and epigenetic mechanisms. The relevance of the molecular mechanisms is discussed in light of current bioavailability data regarding the activity of polyphenols in prostate cancer. We also highlight strategies for improving the bioavailability of polyphenols. We hope that this review will lead to further research regarding the bioavailability and the role of polyphenols in prostate cancer prevention and treatment.

## 1. Introduction

In men, prostate cancer is the second most frequent malignancy of solid organs, after lung cancer [1]. Prostate cancer has several stages of progression, which span from indolent to highly aggressive disease [2]. The lifetime incidence of prostate cancer is about 1 in 6, while the mortality rate is only 33 per 1000 people per year [3,4,5]. These observations led to the conclusion that patients with prostate cancer die from competing risks and not from the disease itself [6,7]. According to the reports from several clinical trials [8,9], the main recommendations in the management of the prostate cancer are conservative management in clinically localized prostate cancer and curative treatment in the intermediate and advanced stages of the disease [5]. Normal epithelium of prostate gland consists of several types of cells: luminal, basal, neuroendocrine, and intermediate [10,11,12]. The majority of prostate cancers (95%) originate from luminal (secretory) cells, while only 1–5% have a neuroendocrine origin [13]. With respect to the localization of AR, luminal cells express AR, while basal cells do not express ARs or have only very low levels [10]. Consequently, the first-line treatment of advance prostate cancer is the administration of androgen deprivation therapy (ADT), which results in reduced levels of androgens followed by a shrinking of the prostate [14]. However, in 12–33 months patients develop resistance to ADT with the acquisition of the castrate-resistant prostate cancer (CRPC) phenotype [15]. In addition, ADT is associated with an increase in fat mass and resistance to insulin, processes that in turn are correlated with the occurrence of cardiovascular diseases and diabetes [16,17]. Beside ARs, other nuclear receptors are involved in the pathology of prostate cancer, such as estrogen receptors [13,18]. In prostate tissue both ER isoforms (ERα and ERβ) are expressed, and aggressive prostate cancer is associated with high ERα expression and reduction in ERβ levels [13,19,20]. Results from preclinical studies reported that the administration of selective ER modulators (SERM), such as tamoxifen, reduced the proliferation of prostate cells [21]. Nevertheless, the administration of SERMs in patients with prostate cancer did not lead to favorable results [22]. Moreover, the average survival is less than 24 months if the disease progresses to the CRPC stage in spite of chemotherapeutic drug administration [23] or secondary hormonal therapy [24]. In addition, side effects of radiation therapy in prostate cancer are associated with both acute effects, such as urinary incontinence and fatigue and later toxicities, such as sexual dysfunction, rectal bleeding, or risk of secondary malignancies in the bladder, colon, or rectum [25]. Another challenge in prostate cancer is related to a small population of cells that are able to maintain malignancy and to facilitate spreading of metastatic cells. These are cancer stem cells (CSCs) and represent approximately 1% of the prostate cancer cells [26]. It was proposed that prostate CSCs originate from normal stem cells, which are localized in the basal layer of the prostate epithelium. At the same time, it is known that the cells from basal layer do not express AR, so prostate CSCs survive ADT [27].

Better prognosis in prostate cancer has been associated with a Mediterranean diet (rich in vegetables and low in saturated fat), in correlation with general recommendations regarding lifestyle factors such as reduction in body mass index, intense physical activity, and avoidance of smoking [28,29]. Several epidemiology studies revealed an inverse correlation between chronic diseases and a diet rich in polyphenols [30,31]. Polyphenols are one of the most numerous and widely distributed natural compounds in the plant kingdom. In recent years, they have received tremendous attention due to a wide range of therapeutic effects, such as anti-inflammatory, antioxidant, antiviral, anticarcinogenic, and anti-allergic properties [32]. Their beneficial effects are reflected further in the chemopreventive activities reported in cancer, metabolic, neurodegenerative, or cardiovascular diseases [32,33,34,35]. In addition, in normal prostate cells, dietary polyphenols do not significantly reduce cell viability and are well tolerated [36,37]. Polyphenols are classified as flavonoid or non-flavonoid compounds [38,39,40], according to the number of aromatic rings and the structural elements that bind these rings together [41]. Flavonoids have a C6‒C3‒C6 structural backbone and are subdivided according to their hydroxylation pattern and variations in the chroman ring into flavones, flavonols, flavanones, and flavan-3-ols [32,42]. Isoflavones, anthocyanins, and proanthocyanidins are also members of the flavonoid family. Flavonoids are found in the plant kingdom as free aglycones, glycosides, methoxides, acylated products, or biflavonoids [32]. A selective list of polyphenols, frequently studied in prostate cancer, is shown in Table 1 [43,44,45,46,47,48,49]. However, the main criticism regarding studies with polyphenols is the lack of correlation between the doses applied in cell cultures and their concentrations in the human digestive system [50,51]. In addition, native polyphenol is used in cell culture studies, while polyphenols probably act as their metabolites and not as pure compounds in the human body [52].

To address these issues, our paper will highlight the molecular mechanisms of action and bioavailability of polyphenols in prostate cancer.

## 2. Molecular Mechanisms of Action of Polyphenols in Prostate Cancer

The major sources of information regarding the molecular mechanisms of action of polyphenols in cancer are in vitro experiments associated with in vivo research. For many years, the activity of polyphenols in cells has been explained by their ability to reduce oxidative stress [64,65]. Nevertheless, several lines of evidence support the capacity of polyphenols to interact with hormone receptors, cell membrane receptors, DNA, and small molecules involved in the cell cycle progression or apoptosis [66,67,68]. Since several types of prostate cell lines are presented in this paper, a brief characterization of them is given in Table 2. A detailed characterization of the cell lines and animal models used in prostate cancer studies is available elsewhere [69,70].

### 2.1. Antioxidant and Pro-Oxidant Activity

Interestingly, polyphenols may play both antioxidant and pro-oxidant roles (Figure 1); the antioxidant effect reduces the disequilibrium induced by the oxidative stress, and the pro-oxidant function induces cytotoxicity in cancer cells [87]. In addition, polyphenols are able to hinder neoplastic processes such as proliferation, invasion, metastasis, or angiogenesis [88,89,90].

Polyphenols demonstrated their antioxidant activity through two main pathways: i) acting as radical scavengers in order to prevent the cellular damage produced by reactive oxygen species (ROS) and ii) acting as molecules that prevent the generation of ROS [91]. In case of radical scavenger activity, the phenolic hydrogen from ortho-dihydroxy or catechol group in the B ring of the flavonoids is transferred to alkyl peroxy radicals, and a quinone, a stable compound, is formed [92]. Polyphenols can generate ROS through their metal chelator activity. It was reported that iron is involved in ROS generation, which in turn will induce DNA damage [91]. Catechols and gallols are metal chelators, which bind iron in a ratio of 3:1 in an octahedral geometry and form catecholate and gallate complexes of Fe^3+^ [91]. Reduction in ROS is observed in several prostate cancer cell lines after the administration of polyphenols. In a prostate cancer cell line (PrEC), quercetin reduced ROS levels in a dose-dependent manner and the addition of quercetin with a potent carcinogen (bezo(a)pyrene) reduced ROS levels compared to administration of the carcinogen alone [93]. Cyanidin-3-O-*β*-glucopyranoside, an anthocyanidin, reduced ROS and increased glutathione (GSH) levels in DU-145, but not in an LNCaP prostate cancer cell line, probably due to an increased basal level of GSH, which conferred protection against the treatment [94].

On the other hand, pro-oxidant activity was reported after the administration of gallic acid, which induced generation of ROS in DU-145 and LNCaP prostate cancer cells [95,96]. In a recent study, three human prostate cancer cell lines, LNCaP, DU-145, and PC-3, incubated in the presence of quercetin, were evaluated by flow cytometry for ROS levels. The interpretation of the results demonstrated that quercetin could act as either an antioxidant or a pro-oxidant depending on the cell line. In the DU-145 cell line quercetin functioned as a pro-oxidant molecule, while in the other two cell lines where the basal level of ROS was higher, quercetin acted as an antioxidant molecule [97].

The antioxidant defense system responsible for protecting cells includes a series of antioxidant enzymes, such as catalase, glutathione peroxidase, or superoxide dismutase [98]. Genistein increased the level of glutathione peroxidase in two human prostate cancer cell lines LNCaP and PC-3, followed by an increase in glutathione peroxidase enzyme activity [99]. In a model of hormone- and carcinogen-induced prostate cancer in Sprague‒Dawley rats, supplementation of quercetin during prostate cancer initiation resulted in increased levels of antioxidant enzymes superoxide dismutase (SOD), catalase (CAT), glutathione peroxidase (GPx), and glutathione reductase (GSR) compared to the control group without quercetin supplementation [100]. However, data reported in 2001 by Salganik and in 2011 by Thomas and colleagues suggested that the consumption of epigallocatechin gallate (EGCG) may interfere with radiotherapy, which is regularly applied in localized prostate cancer. The proposed mechanism involved increased levels of SOD enzymes (manganese superoxide dismutase/MnSOD and CuZnSOD), which in turn will clear the ROS required for induction of apoptosis after exposure to radiotherapy [101,102].

Dietary polyphenols displayed both antioxidant and pro-oxidant activities in prostate cancer cells with the final aim of reducing the malignant phenotype through decreasing ROS levels, increasing antioxidant enzyme activity, or inducing cytotoxicity. Nevertheless, caution should be exercised in the combined administration of dietary polyphenols and radiation therapy, since the natural compounds may scavenge ROS, decreasing the therapeutic effect of radiation.

### 2.2. Proliferation and Survival

#### 2.2.1. Androgen receptors

The differentiation and development of the male urogenital system require the presence of androgens and their receptors [103]. However, besides their beneficial role, androgens and androgen receptors are also involved in the proliferation of prostate cancer cells [104]. The first therapeutic approach proposed in advanced prostate cancer is based on ADT. Nevertheless, some prostate cancers become “androgen refractory” during ADT, probably due to mutations or amplification of the AR gene [105]. In human prostate cancer, three main mechanisms are proposed to explain the anti-androgenic effects of polyphenols: direct competition, inhibition of the activity of androgen receptors, and inhibition of transactivators of androgen receptor [106]. The direct competition between polyphenols and androgens appears to be based on their structural similarities (Figure 2).

The concentration of the polyphenols, particularly of genistein, appears to be a critical factor in prostate cancer cells. Physiological doses of genistein increase the levels of AR, whereas higher doses induced inhibitory effects, demonstrating a biphasic effect [107]. Resveratrol inhibited the expression of AR in prostate cancer cells [108,109,110,111] and its activity was synergic with flutamide, an antagonist of AR [112]. EGCG was reported to physically interact with the ligand-binding domain of AR, which in turn decreased the transcriptional activity of AR and consequently reduced the growth of the prostate cancer cells [113]. Furthermore, EGCG reduced AR expression by 20% and 30% at the mRNA and protein levels, respectively, in androgen-dependent prostate cancer cells [114]. Similar results were reported in a xenograft mouse model for prostate cancer, where EGCG inhibited the nuclear translocation of AR and its downstream effects [113].

#### 2.2.2. Growth Factors and Cytokine Receptors

In certain cellular signaling events ARs can cross-talk with the pathway of growth factors (GF) [115]; this communication between ARs and GFs might be responsible for the acquired resistance after ADT in prostate cancer [116]. Administration of apigenin to transgenic adenocarcinoma of the mouse prostate (TRAMP) mice induced a reduction in the levels of insulin-like growth factor 1 (IGF-1) levels [117]. In prostate cancer cell lines, resveratrol reduced the levels of several receptor tyrosin kineses (EGFR, ErbB2/HER2, IGFR-1) in a time-dependent manner, particularly in androgen-negative prostate cancer cells [79,118]. The level of pro-inflammatory chemokine (C-X-C motif) ligand -1, -2 (CXCL-1, and -2) was reduced by curcumin in PC-3 prostate carcinoma cells [119] and the combined administration with β-phenylethyl isothiocyanate, an apoptotic inducer extracted from cruciferous vegetables, reduced phosphorylation of p-EGFR (Tyr 845, Tyr 1068) [120]. EGCG inhibited the auto-phosphorylation of c-Met/hepatocyte growth factor receptor (HGFR) at Tyr1234/1235 and modified the structure of lipid rafts in DU145 prostate cancer cells [121].

#### 2.2.3. Signal Transduction

The signaling pathways are complex and, in the course of tumor progression (Figure 3), cancer cells acquire characteristics that allow them to survive and proliferate in a non-regulated manner [122,123,124]. Apigenin decreased the expression levels of nuclear factor kappa-light-chain-enhancer of activated B cells (NF-κB), phosphatidylinositol 3-kinase (PI3K), proteinkinase B (PKB/Akt), in androgen-negative prostate cancer cells and in CD44^+^ cancer stem cells isolated from the same cell lines [82]. Furthermore, apigenin inhibits cancer progression, targeting phosphorylated PI3K, Akt, extracelluar signal-regulated kinases 1/2 (ERK 1/2), and forkhead box class O (FoxO) in the dorsolateral prostate of TRAMP mice [117,125].

Resveratrol lowered PI3K, Akt, and glycogen synthase kinase-3 (GSK-3) activities in androgen-dependent prostate cancer cells [108] and reduced the total and phosphorylated levels of PI3K, Akt, and phosphatase and tensin homolog (PTEN) in both androgen-positive and -negative prostate cancer cells [118]. Caffeic acid phenethyl ester (CAPE), a natural compound from propolis, reduced cell proliferation and colony formation in the PC-3 human prostate cancer cell line [130]. The proposed mechanisms include a reduction in phosphorylation of p-ERK1/2 (Thr202/Tyr204), p-Akt (Ser473), p-mTOR (Ser2448, Ser24981), p-GSK3α (Ser21), and p-GSK3β (Ser9). In addition, CAPE displayed a synergistic suppressive activity in combination with vinblastine and paclitaxel [130]. In prostate cancer xenografts in nude mice, CAPE inhibited tumor growth; the proposed mechanisms include reduction in the level of signaling molecules from the Akt pathway [131]. The levels of NF-κB, a nuclear factor required for cell survival, are reduced by curcumin in PC-3 and LNCaP prostate cancer cells [119,132]. Combinatory administration of curcumin and β-phenylethyl isothiocyanate reduced the phosphorylation of p-PI3K, p-Akt (Ser 473, Thr 208), and p-IκBα in PC3 cells [120]. In prostate cancer cells, gallic acid reduced the levels of the following proteins: son of sevenless homolog 1 (SOS1), growth factor receptor bound protein 2 (GRB2), protein kinase C (PKC), NF-κB, c-Jun N-terminal kinase (JNK), ERK1/2, p38-MAPK, and p-Akt (Thr308, Ser 473) [133]. Gingerol reduced multidrug resistance-associated protein 1 (MRP1) in PC3 cells [134], while resveratrol reduced the levels of ERK1/2 and Akt [79]. EGCG inhibited the activation of NF-κB, reduced the phosphorylation of ERK1/2, and lowered HFG-induced phosphorylation of p-Akt (Ser473), p-ERK (Thr202/Tyr204), but not p38-MAPK (Thr180/Tyr182) in DU-145 cells [90,121].

The main mechanism of action confirms direct competition between polyphenols and DHT for AR binding, particularly in the case of EGCG. Other mechanisms of action of polyphenols include the inhibition of key molecules involved in alternative activation pathways of AR in prostate cancer, such us PI3K/Akt, NF-κB, or MAPK. Nevertheless, besides the cell-line-dependent results (androgen-positive or androgen-negative models), the concentration of polyphenols plays a central role in controlling the level of AR. In particular, genistein at physiological levels increased the levels of AR, while a higher concentration resulted in decreased levels of AR. Taken together, these studies support the hypothesis that polyphenols are able to modulate AR and its transactivation pathways in prostate cancer models, both in cell culture and in animal experiments.

### 2.3. Cell Cycle and Apoptosis

#### 2.3.1. Cell Cycle

In mammalian cells, the cell cycle is tightly regulated by a series of protein complexes and checkpoints. The complexes formed by cyclins and cyclin-dependent kinases (CDKs) drive the progression of the cell cycle through four phases, namely mitosis and DNA synthesis separated by two gap phases, G1 and G2 (Figure 4). While the majority of cells in the human body do not enter the cell cycle, in malignancies cells lose control over the progression of the cell cycle and proliferation [135,136,137,138]. Several genetic modifications, such as gain-of-function mutations in proto-oncogenes, gene amplification, or loss-of-function mutations of tumor suppressor genes are considered responsible for the acquisition of malignant phenotype [135].

Apigenin reduced the level of cyclin D1, followed by cell cycle arrest in the G0/G1 phase in LNCaP and PC-3 prostate cancer cell lines [125]. Nevertheless, the effect of apigenin on the progression of cell cycle is cell-line-dependent, since in another prostate cell line (DU-145), the flavonoid arrested the cells in G2/M phase [139]. In PC-3 prostate cancer cells, CAPE decreased cyclin D1 and cyclin E, but not p21 protein expression [130], while cyanidin-3-O-*β*-glucopyranoside induced p21 protein expression in the DU-145 and LNCaP prostate cancer cell lines [94]. Gallic acid reduced the expression of cell division cycle protein 2 (cdc2)/CDK1, CDK2, CDK4, CDK6, cyclin B1, and cyclin E in prostate tissue from TRAMP mice, in correlation with a decreased proliferative index [140]. In addition, gallic acid induced the arrest of cells in the G2/M phase through the inhibition of cell cycle division 25 (cdc25) phosphatase and activation of checkpoint kinases 1 and 2 (CHK1 and CHK2) in DU-145 prostate cancer cells [95]. Combined administration of gallic acid and doxorubicin demonstrated synergistic effects in suppressing DU-145 prostate cell growth [95]. Gingerol affected the progression of the cell cycle in correlation with proliferation index both in PC-3 prostate cancer cell lines and in xenograft mice for prostate cancer [71]. In PC-3 cells, gingerol reduced the levels of cyclin D1, cyclin E, and CDK4 in correlation with a decreased proliferative index. In vitro results have been supported by in vivo effects in xenograft tumors [71]. Hesperidin suppressed proliferation in both androgen-dependent and androgen-independent prostate cancer cell lines [141,142]. In AR-positive cancer cell lines, resveratrol induced a biphasic effect on DNA synthesis—physiological concentration of resveratrol increased DNA synthesis, while higher concentrations inhibited DNA synthesis [143]. Microarray assay demonstrated the inhibition of genes involved in cell cycle progression after incubation with resveratrol [67]. Administration of Polyphenon E, a combination of major catechins from green tea, induced cell cycle arrest in G0/G1 in PNT1, a model cell line for the initial stages of prostate cancer and in G2/M in PC-3, a model cell line for advanced stages. In correlation with the previous results, cell viability was highly dependent on the cell line: PNT1 cells responded to lower doses of Polyphenon E compared to its advanced counterpart [144]. Quercetin induced G0/G1 arrest and sub-G1 accumulation of PC3 cells in connection with decreased levels of CDK2, cyclin E, and cyclin D [68].

#### 2.3.2. Apoptosis

Maintenance of tissue integrity, embryonic development, or proper function of the immune system is based on programmed cell death or apoptosis [145,146,147]. Besides the self-destructing processes required for maintaining tissue homeostasis, apoptosis is essential for the inhibition of the malignant pathology [148]. There are three major apoptotic pathways (Figure 5): (i) an extrinsic pathway based on activation of the cell death receptors in conjunction with caspases-8 and -10; (ii) an intrinsic pathway focused on the processes triggered by activated mitochondria; and (iii) a granzyme pathway characteristic of immune cells [145]. The intrinsic pathway involves the activation of pro-apoptotic members of B-cell lymphoma 2 (Bcl-2) proteins, such as Bcl-2-associated X protein (Bax) and Bcl-2 homologous antagonist/killer (Bax), which in turn will induce the release of cytochrome *c* in the cytoplasm with the formation of the apoptosome and activation of executioner caspases [147]. The proposed mechanisms contributing to the circumvention of apoptosis and induction of malignancy may include impaired cell death receptor activity, defects in tumor suppressor gene *TP53*, reduction in the expression of caspases, or loss of the equilibrium between pro- and anti-apoptotic Bcl-2 proteins [149]. Apigenin induced collapse of the mitochondrial membrane potential, followed by the release of cytochrome *c* into the cytoplasm, decreased the levels of anti-apoptotic proteins Bcl-2 and Bcl-2-extra-large (Bcl-XL) proteins, and increased the level of Bax [150]. Moreover, the apoptotic processes produced by apigenin have been demonstrated by induction of the elevated levels of TNF-related apoptosis-inducing ligand (TRAIL) and death receptor 5 (DR5) in prostate cancer cells [150,151]. In addition, apigenin upregulated the level of caspase-3 and -8 in cancer stem cells isolated from androgen-negative prostate cancer cells [82]. Cyanidin-3-O-*β*-glucopyranoside activated caspase-3 and induced DNA fragmentation in several prostate cancer cell lines [94,150]. Treatment of the LNCaP prostate cancer cell line with curcumin increased the sensitivity of cells to TRAIL-induced apoptosis and induced the externalization of phosphatidylserine, a marker for early apoptosis [132]. The cytotoxic effects and reduced viability triggered by curcumin have been observed in a metastatic prostatic cancer cell line obtained from an invasive tumor with a Gleason score of 9 [152]. Gallic acid induced apoptosis through the collapse of the mitochondrial membrane potential, release of cytochrome *c*, and activation of caspase-3, -8, and -9 [96].

Gingerol might induce apoptosis both in vitro, using PC-3 prostate cancer cells, and in vivo, using mice with prostate cancer xenografts, by increasing the level of cleaved Poly (ADP-ribose) polymerase (PARP) and reducing the levels of anti-apoptotic protein Bcl-2 [71]. EGCG, the major catechin from green tea, reduced the level of Bcl-2, increased the level of Bax, and induced the activation of caspase-3, -8, -9, and PARP in a dose- and time-dependent manner in androgen-positive prostate cancer cells [154]. Similar effects have been observed after the administration of Polyphenon E in two cell lines that are models for the initial and advanced stages of prostate cancer [144]. However, since the authors noticed the enlargement of the endoplasmic reticulum, upregulation of CCAAT-enhancer-binding proteins homologous (CHOP), and p53-upregulated modulator of apoptosis (PUMA) proteins, they suggested that cell death is necroptosis rather than apoptosis [144]. Combination of EGCG with ibuprofen, an anti-inflammatory drug, or Zn^2+^, enhanced the apoptotic effects induced by the polyphenol itself [155,156]. Nevertheless, in an attempt to evaluate whether the zinc‒EGCG complexes are more effective than free Zn^2+^ and EGCG in inducing membrane permeability effects, the results indicated that only free compounds were effective, while the complexes were not [157]. Quercetin induced apoptotic effects, characterized by reduced levels of Bcl-2, a collapse of the mitochondrial membrane potential, and activation of caspases-3, -8, and -9 in PC-3 cells [68]. Nevertheless, similar to EGCG administration, quercetin elevated the levels of markers involved in ER stress, such as activating transcription factor (ATF), 78-kDa glucose-regulated protein (GRP78), and growth arrest- and DNA damage-inducible gene 153 (GADD153) [68]. In order to increase its solubility, quercetin can be administrated as nanomicelles [158]. Comparing quercetin and nanomicelles loaded with quercetin, the latter demonstrated better results in terms of the reduction of cell viability, inhibition of colony formation, and increased level of morphological changes. In addition, quercetin-loaded nanomicelles increased the number of early and late apoptotic cells, reduced the mitochondrial membrane potential, reduced Bcl-2, and increased Bax levels. In line with previous results, treatment with nanomicelles loaded with quercetin increased caspase-3 levels in prostate cancer from xenografted mice and significantly reduced the tumor size [158].

In experimental models of prostate cancer, polyphenols inhibit most of the steps responsible for the loss of control over cell cycle progression. The dietary compounds reduced the levels of several members belonging to cyclin‒CDKs complexes, increased the expression level of proteins coded by tumor suppressor genes, and increased the levels of checkpoint proteins. Coordinating all these events resulted in cell cycle arrest in G0/G1 or G2/M phases depending on the compound and the cell line model applied. The effects of polyphenols in cell cycle experiments are completed with apoptosis, another cellular process often dysregulated in cancer cells. The main effects of polyphenols on apoptotic events include activation of caspases (caspase-3,-8-9) and increase in the level of pro-apoptotic proteins (Bax, Bak), death cell ligands, and their receptors (TRAIL, DR5) in conjunction with a reduction in the expression of anti-apoptotic proteins (Bcl-2). Nevertheless, during treatment with polyphenols, events associated with the endoplasmic reticulum suggested that cell death induced by the dietary compounds is not restricted to apoptosis, but might include necroptosis as well.

### 2.4. Invasion, Metastasis, and Angiogenesis

#### 2.4.1. Invasion and Metastasis

In an excellent review published by Valastyan and Weinberg in 2011, the invasion‒metastasis cascade is described in detail [159]. Briefly, the major steps include local invasion of the cells with origin in the primary tumor, intravasation into the blood stream, survival in the blood flow, arrest in the distant organ, extravasation, micrometastasis formation, and metastatic colonization [159,160]. A main feature of metastasis is epithelial‒mesenchymal transition (EMT), which might be activated by a large series of factors such as extracellular matrix (ECM) structural molecules, growth factors, and extracellular proteases. The literature data showed that the most studied extracellular proteases involved in invasion are matrix metalloproteinases (MMP) or urokinase plasminogen activators (uPA) [161]. EMT has its inverse process during the colonization of distant organs, entitled mesenchymal‒epithelial transition (MET), when all processes required for invasion are reversed [162]. During prostate cancer progression the level of enzymes involved in metastasis, uPA, MMP-2, and MMP-9, displayed increased levels in the dorsolateral prostate of TRAMP mice, but the intake of apigenin downregulated their levels, which led to a “complete absence” of the metastasis [117]. In prostate cancer cells, apigenin reversed the EMT phenotype by increasing the level of E-cadherin, a marker for epithelial cells, and reducing the level of vimentin, a mesenchymal indicator [139]. Curcumin reduced the levels of chemotactic cytokines CXCL-1 and -2, cyclooxygenase (COX), secreted protein acidic and cysteine-rich (SPARC)/osteonectin and EGF-containing fibulin-like extracellular matrix protein 1 (EFEMP) proteins in PC-3 prostate cancer cells [119]. Moreover, the number of lung metastases is reduced after the administration of curcumin in an orthotopic mouse model for hematogenous metastasis with PC-3 cells [119]. EGCG and gallic acid suppressed the invasion and migration of prostate cancer cells in correlation with reduced levels of MMP-2 and -9 [90,133]. Genistein reduced the level of MMP genes or proteins in several prostate cancer cells and the proposed molecular mechanism is based on a reduction of the phosphorylation of heatshock protein 27 [163,164,165]. In addition, the effects of genistein have been evaluated in mouse models. In the case of mouse models for bone metastasis with prostate cancer cells, genistein reduced the level of MMP-9 [88], while in an orthotopic model of prostate cancer it inhibited metastasis to lymph nodes and lungs [166]. The number of metastatic lung lesions decreased after the administration of resveratrol in a xenograft mouse model of prostate cancer obtained from an aggressive AR-negative cancer cell line [79].

#### 2.4.2. Angiogenesis

Formation of novel blood vessels or angiogenesis, one of the major steps in cancer, enables the development of the tumor by providing nutrients and oxygen [167]. Furthermore, the transport of malignant cells from the primary tumor to the distant organ is favored by blood vessels [168]. It is known that endothelial cells might be modified from a quiescent phase to a growing one under the action of angiogenesis regulators [169]. The list of angiogenesis regulators might include vascular endothelial growth factor (VEGF), transforming growth factor beta (TGF-β), interleukin-8 (IL-8), fibroblast growth factor (FGF), platelet-derived growth factor (PDGF), angiopoietins, and so on [170]. Several models of tumor angiogenesis are proposed, such as sprouting angiogenesis, vascular mimicry, vessel splitting, and others [167]. Despite the high interest in identifying potential molecules involved in prevention of the angiogenesis, studies on the effect of polyphenols on the molecular mechanisms of angiogenesis in prostate cancer are very rare. Only a few papers described the role of apigenin, genistein, quercetin, or green tea polyphenols in reducing the levels of VEGF in prostate cancer [117,164,171,172]. Quercetin inhibited VEGF secretion in PC-3 prostate cancer cells through the activation of the AKT/mTOR/ribosomal protein S6 kinase (p70-S6K) pathway; in addition, in the same set of experiments quercetin inhibited tube formation of the human umbilical vein endothelial cells (HUVEC) cells in parallel with reduced phosphorylation of p-VEGFR2 (Tyr1175) [172].

Promising results have been reported after the administration of polyphenols in mouse models for metastasis associated with prostate cancer. The major observation includes a reduction in the number of lung metastasis after ingestion of polyphenols in mouse models for prostate cancer. The polyphenols responsible for these results were curcumin, genistein, or resveratrol. Even more, the data reported with respect to apigenin administration indicated a lack of metastasis in a transgenic mouse model for prostate cancer. The proposed molecular mechanisms responsible for the decline of metastatic tumors include reduction of the enzyme levels involved in invasion processes, such as MMPs, uPA, increase of the EMT markers, such as E-cadherin, or decrease of the MET markers, such as vimentin. Regarding the anti-angiogenesis data, there have only been a few reports that several polyphenols reduced the level of VEGF in prostate cancer models [117,164,171,172]. These data suggest that more experiments are needed to further elucidate the modalities to inhibit new vessel formation in prostate cancer.

### 2.5. Genomics and Epigenomics

#### 2.5.1. Oncogenes

The activity of oncogenes was initially identified in virus-induced avian or murine sarcomas [173]. Gain-of-function mutations, described in the structure of proto-oncogenes, transform them into oncogenes with increased ability to stimulate or enhance cell division and proliferation [174]. The oncogenes can be divided into four major categories: growth factors (*v-sis*, coding for PDGF), growth factor receptors (*v-erbb*, coding for EGFR), signal transduction proteins (*v-ras*, coding for rat sarcoma protein/Ras protein), and transcription factors (*v-jun*, identified in avian sarcoma 17; *v-fos*, identified in FBJ murine osteosarcoma; *v-myc*, identified in avian myelocytomatosis) [173]. Experimental data indicated that the inactivation of one oncogene might reverse the malignant phenotype; this may represent the Achilles’ heel since the cancer is “addicted” to the activation of the oncogenes [175,176]. Again, data regarding the prostate cancer, oncogenes, and polyphenols are scarce. In androgen-dependent prostate cancer cells, curcumin inhibited the phosphorylation of c-Jun protein at Ser73 [177], while the same transcription factor was inactivated by EGCG in androgen-negative prostate cancer cells [90].

#### 2.5.2. Tumor Suppressor Genes

Inhibition of the alterations occurring during oncogenesis might be achieved by the activation of tumor suppressor genes [178]. The most studied tumor suppressor genes include *TP53*, *RB*, breast cancer susceptibility gene (*BRCA*), and *PTEN* [178,179]. However, their protection can be interrupted by a loss of heterozygosity mutation [178]. Apigenin stabilizes tumor suppressor protein p53 by phosphorylation of alternate frame reading protein (p14ARF) and upregulation of p27 protein in prostate cancer cells [125,150]. It was reported that curcumin increased the expression level of p53 in prostate cancer cells from lung metastasis in a mouse model [119], while EGCG increased the levels of p53 and p21 in a dose- and time-dependent manner in androgen-dependent prostate cancer cells [154].

#### 2.5.3. DNA Methylation and Histone Modification

Epigenetic mechanisms involve the modification in the gene status by activating or silencing the transcription, without changes in the DNA sequence [180]. The phenomenon is extremely complex due to the high diversity of genomic DNA [181]. However, the major biochemical mechanisms related to epigenetic modifications might be summarized as methylation, acetylation, phosphorylation, or ubiquitination [180,181]. Hypomethylation is correlated with genome instability, activation of transposons and proto-oncogenes, while hypermethylation might silence genes involved in anticancer mechanisms, such as tumor suppressor genes or genes involved in promoting apoptosis or cell cycle arrest [182]. For instance, in prostate cancer the transposable elements Alu (DNA sequence first identified with restriction endonuclease isolated from *Arthorbacter luteus*) and long interspersed nuclear elements 1 (LINE-1) have been shown to be hypomethylated, while CpG island loci are hypermethylated [183]. The results regarding the influence of polyphenols on the epigenetic mechanisms in prostate cancer cells indicate that some of the compounds might influence DNA methylation or histone acetylation status, while others do not. For instance, in PC-3 prostate cancer cells curcumin prevents histone 3 (H3) hyper-acetylation and inhibits the acetyltransferase activity of p300, a member of the histone acethyltransferases (HATs) group [184]. Genistein decreased the level of methylated and increased the level of un-methylated retinoic acid receptor β (RARβ) in a dose-dependent manner in LNCaP and PC-3 prostate cancer cells [185]. The activation of a tumor suppressor gene, B-cell translocation gene 3 (*BTG3*), which is downregulated in cancer due to its hypermethylation, was initiated by genistein through the demethylation of CpG island in the promoter of the gene in prostate cancer cells [186]. Vardi and colleagues proposed the demethylation of CpG islands from the promotor of tumor suppressor genes by genistein as a protection mechanism against the progression of prostate cancer [187]. However, the evaluation of 5-methyl-deoxycytidine levels, the methylation of B1 repetitive element and Mage-a8 gene promoter, at foci that are hypermethylated in TRAMP mice did not provide any evidence that green tea polyphenols are inhibiting their epigenetic changes [188].

#### 2.5.4. miRNA

Since 1993, when microRNA (miRNA) was discovered by two research groups, the activity of miRNA has been intensely studied as biomarkers or possible targets in cancer therapy [189,190]. miRNAs are small non-coding RNA molecules (18–25 nucleotides), which in the majority of cases interact with the 3′ or, more rarely, the 5′ untranslated regions (UTR) of the mRNA target and exert both oncogenic or tumor suppressor effects [191,192]. The oncogenic miRNAs that are upregulated in prostate cancer may include miR-21, miR-222, miR-32, and miR-125b, while examples of suppressive miRNAs downregulated in prostate cancer are miR-34a, miR-145, miR224, and miR-133 [193]. As an alternative to systemic chemotherapy, after the acquisition of CRPC, Siddiqui and his colleagues propose administration of EGCG as an inhibitor of the oncogenic miR-21 and upregulator of tumor suppressor miR-330 [113]. The increased expression of the oncogenic miR-151 in prostate cancer cells was downregulated after incubation with genistein [73]. At the same time, genistein increased the expression of tumor suppressor miR-574-3p in prostate cancer cells [73]. Combination of quercetin with hyperoside, a phenolic compound, reduced the overexpression of oncogenic miR-21 in prostate cancer cells [194]. Resveratrol decreased metastatic lung lesions in a severe combined immunodeficient (SCID) mouse xenograft model of prostate cancer obtained from an aggresive AR-negative prostate cancer cell line and the proposed mechanism of action includes the reduction of oncogenic miR-21 [79].

Since the hypothesis of cancer getting addicted to their highly active oncogenes has already been proposed, chemopreventive or therapeutic approaches should be planned to target these genes. However, very few data are reported in respect to the activity of polyphenols on oncogenes in prostate cancer. Interestingly, the activity of the proteins coded for by the oncogenes can be inhibited by the polyphenols in both types of prostate cancer, androgen-positive and androgen-negative, without the limitation imposed by the presence of AR. In addition, curcumin and EGCG increase the level of p53. Genistein prevents histone acetylation and decreases the hypermethylation of the tumor suppressor genes, while the reported results do not support EGCG as a modulator of epigenetic changes in prostate cancer. Nevertheless, encouraging results are coming from a recent field of miR research. Oncogenic miRs are suppressed by EGCG, genistein, and resveratrol, whereas the expression of tumor suppressor miR is increased by EGCG and genistein in models of prostate cancer. Since the data regarding the influence of polyphenols on genetic and epigenetic modifications in prostate cancer are very few, for a better understanding of the biology of the disease, additional studies will be helpful. A summary of the molecular targets of polyphenols in prostate cancer is presented in Figure 3, Figure 4 and Figure 5 and Table 3.

## 3. Bioavailability of Polyphenols in Prostate Cancer

Biological properties of polyphenols strongly depend on their bioavailability [199]. According to the U.S. Food and Drug Administration (FDA) bioavailability is defined as “the rate and extent to which the active ingredient or active moiety is absorbed from a drug and becomes available at the site of action” [200,201]. Bioavailability of phenolic compounds from different sources involves several steps: liberation from the food/medicinal plant matrix (also known as bioaccessibility), absorption, distribution, metabolism, and elimination [202,203]. The bioavailability of the phenolic compounds is relatively low due to poor absorption, extensive biotransformation, and rapid clearance from the body [202,203]. Polyphenol bioavailability is influenced by several factors such as chemical structure, dietary intake, food matrix, food processing, and interaction with other food components or time of harvest for plants. Host-related factors, such as gastrointestinal absorption, metabolism of polyphenols, plasma transport distribution, and elimination [54,55,201] are of great importance. Excellent reviews describe the bioavailability of polyphenols in humans [55,204]; this is beyond the aim of the present paper. Furthermore, we will focus on the effect of polyphenols in prostate cancer, particularly on the role of polyphenols’ metabolites in prostate cancer, together with strategies for enhancing polyphenol bioavailability.

### 3.1. Role of Polyphenol Metabolites in Prostate Cancer

Recent research has demonstrated that, although polyphenol bioavailability is relatively low, polyphenols still exhibit biological functions, mainly due to their bioactive metabolites [205]. Hepatocytes and enterocytes are involved in polyphenol modifications through phase I and phase II metabolizing enzymes [199,206]. The phase I reactions involve cytochrome P450 enzymes (CYP450), which modify the polyphenol structure to facilitate phase 2 conjugation reactions [199,206]. Phase I reactions usually include oxidation, reduction, and hydrolysis, which serve to increase the hydrophilicity of the molecule and expose or add certain functional groups (such as hydroxyl groups) that will facilitate phase II reactions [199,206]. Most polyphenols, unlike drugs, are not substrates for CYP 450 enzymes, but after deconjugation (by lactase phloridzin hydrolase and cytosolic β-glucosidase enzymes), they undergo phase II reactions [202].

Several studies have outlined the importance of polyphenol metabolites in prostate cancer [43,44,48,49,207,208,209]. Moreover, recent studies have demonstrated that metabolites of ellagitannins (urolithins, elagic acid) and flavan-3-ols (3′, 4′, 5′–trihydroxyphenyl)–γ-valerolactone) are widely distributed in the prostate [210,211]. Metabolites of isoflavones, ellagitannins, anthocyanins, proanthocyanidins, and flavan-3-ols (Table 4) are usually generated by colonic microbiota [205,212,213,214,215,216,217,218,219], while resveratrol conjugates are formed under the influence of phase II metabolizing enzymes (uridine diphosphoglucuronosyl transferases/UGTs, sulphotransferases/SULTs, glutathione-S-transferases/GSTs). These enzymes are found in the liver and enterocytes [202,206]. Resveratrol conjugates undergo enterohepatic circulation, which is responsible for a longer exposure of the body to resveratrol and a longer half-elimination time (t_1/2_) [220].

According to recent research, ellagitannin metabolites (ellagic acid and urolithins) inhibit androgen-independent proliferation of prostate cancer cells due to their ability to induce cell cycle arrest in the G2/M phase and inhibit the cyclin B1/cdc2 complex, which is a key regulator in the initiation of mitosis [209]. Moreover, ellagic acid decreases the expression of cyclin D1, which promotes cell cycle progression [209] while urolithins induce apoptosis with a decrease in Bcl-2 proteins and attenuate the function of the androgen receptor by repressing its expression, causing a downregulation of prostate-specific antigen (PSA) levels [46]. An association between urolithin A and green tea catechin metabolites (3′, 4′, 5′–trihydroxyphenyl)–γ-valerolactone) had synergistic antiproliferative effects on a human androgen-dependent LNCaP prostate cell line, inhibited PSA secretion, and induced proliferation via DHT [221]. It was shown that ellagic acid caused a significant decrease in the activity of heme oxygenase system (OH1, OH2), which is a potent regulator of cell growth and angiogenesis. Ellagic acid also caused significant inhibition of vascular endothelial growth factor (VEGF) activity and osteoprotegerin (OPG). OPG is a marker of bone metastasis, frequently associated with prostate cancer [222].

Enterolactone and enterodiol, mammalian phytoestrogens that are derived from lignans and are produced in the colon under the action of gut microbiota, impede tumor cell proliferation in men with localized prostate cancer through decreased expression of VEGF. Moreover, urinary concentrations of enterolactone were significantly and inversely correlated with Ki-67 (a marker of cell proliferation) in prostate tumor tissue [44]. According to recent research, enterolactone restricted the proliferation of the middle and late stages of prostate cancer through increased expression of the PTEN tumor suppressor gene and reduced expression of the miR-106b cluster (miR-106b, miR-93, and miR-25) [223]. A growing body of epidemiological studies has revealed that the incidence of prostate cancer is reduced in men with a diet rich in isoflavones, and equol-producing intestinal microflora carry a significantly reduced risk [224].

Possible mechanisms involved in the antitumor activity of isoflavone and equol are downregulation of genes involved in cell cycle regulation [226], selective activation of the estrogen receptors ERβ in the prostate [227], increased expression of transcription factors with tumor suppressor activity [49], and suppression of androgen-receptor signaling [48]. Another important metabolite of genistein and daidzein, O-desmethyl angolensin, showed antiproliferative effects towards benign prostate epithelial cells and LNCaP cancer cells [225]. Protocatechuic acid, a major anthocyanin metabolite, has shown apoptotic effects in prostate cancer (LNCaP) cells through enhanced DNA fragmentation, reduced mitochondrial membrane potential, elevated caspase-3 activity, suppressed VEGF activity, and lowered levels of pro-inflammatory agents (IL-6, IL-8) [43]. Flavan-3-ols (catechin, epicatechin) and proanthocyanidin metabolites such as 3-hydroxyphenyl propionic acid, 4-hydroxyphenylacetic acid, and hippuric acid also exhibited anti-proliferative activity in prostate cancer cells [207]. Regarding stilbenes, resveratrol conjugates (mainly sulfates) contribute to the anticancer effects of resveratrol by delivering resveratrol to tissues in a stable conjugated form, enabling gradual regeneration of the parent compound within the cells [208]. Some metabolites of polyphenols play an important role in the protection against prostate cancer and act as antitumor agents through various mechanisms such as cell cycle arrest, inhibition of angiogenesis, downregulation of PSA levels, selective activation of ERβ prostate receptors, suppression of androgen receptor signaling, induction of apoptosis, or anti-inflammatory effects.

### 3.2. Strategies for Enhancing Polyphenol Bioavailability

It is well known that polyphenol bioavailability is low due to poor absorption, low permeability and solubility, instability under conditions encountered in the gastrointestinal tract (pH, enzymes, presence of other nutrients), insufficient gastric residence, or interaction with other dietary compounds (milk, carbohydrates, or phytic acid from fibers) [202,203,205,228,229,230,231]. Moreover, some of the previously mentioned compounds (genistein, resveratrol, EGCG) are substrates for breast-cancer-resistant proteins (BCRP) and P-glycoprotein (P-gp), which are active transporters that limit the gastrointestinal absorption of their substrates [206]. Flavan-3-ols (catechin, epicatechin, EGCG) are quickly eliminated from the body, showing decreased t_½_ (less than one hour) [232,233]. Regarding the pharmacokinetic profile of oligomeric proanthocyanidins, polymerization impairs their intestinal absorption, since dimers and trimers were found in small concentrations in both human/rat plasmas one or two hours post-administration [204,234,235]. Bioavailability studies with resveratrol, gingerols, and elagic acid (generated after acid hydrolysis of ellagitannins) have also shown rapid metabolism, with low plasma concentrations [213,236,237,238,239]. Very low plasma concentrations were also observed for secoisolariciresinol metabolites (enterodiol, enterolactone) [240]. For other polyphenols (such as anthocyanins), more than 50% of activity is lost due to the enzymatic action of oral microbiota [15,230].

Considering all these aspects, the development of new strategies to enhance polyphenol bioavailability is a subject of present interest. The main strategies applied to enhance polyphenol biodisponibility are drug delivery systems such as lipid nanoparticles/nanostructured lipid carriers, nanocrystals, liposomes, emulsions, niosomes, micelles, cyclodextrins, implant delivery systems, or food macromolecule based delivery [224,228,241,242]. Additionally, other systems have been used to augment the biodisponibility of polyphenols, such as solid dispersion techniques for microencapsulation, de novo formulations, or alternative routes of administration [243,244].

The most common carrier agents used for the micro/nano-encapsulation of bioactive compounds are polysaccharides (starch, dextrins, maltodextrins, cyclodextrins), celluloses (methylcellulose, carboxymethyl cellulose), pectins, chitosan, and polylactic acid‒polyethylen glycol [245,246,247]. Solid lipid nanoparticles (SLNs) and nanostructured lipid carriers (NLCs) have properties of controlled release, subcellular size, avoidance of organic solvent, and biocompatibility with tissues and cells. They can be used for improving solubility or the pharmacokinetic profile of phytochemicals or drugs [241,247]. Liposomes have a biphasic character, so can serve as carriers for both hydrophilic and hydrophobic compounds [247], while emulsions, and micelles increase the aqueous solubility of hydrophobic compounds [247]. Niosomes resemble liposomes but are non-ionic and less toxic; they increase the stability of the entrapped drug and their action is restricted to target cells and tissues [241]. Nanosized carriers loaded with drugs have proven therapeutic benefits for different types of cancers, due to higher antitumor efficiency, specific delivery of drug to the target organ, greater cytotoxicity, and inhibition of P-glycoprotein-dependent multidrug resistance [248]. Nanosized drug carriers have been used for improving the bioavailability of EGCG [242], resveratrol [249], flavones and flavonols (quercetin, luteolin, apigenin) [245,250,251], and curcumin [252].

Another approach for increasing the bioavailability of polyphenols is the use of cyclodextrins, which are unique molecules with a pseudo-amphiphilic structure. They enhance bioavailability through increasing drug solubility and dissolution [241]. Cyclodextrins have been used to improve the bioavailability of flavonoid (rutin, chrysin, baicalein, myricetin, apigenin, genistein, daidzein) and non-flavonoid (chlorogenic acid, ferulic acid, caffeic acid, resveratrol) compounds [253]. Implant delivery systems use non-degradable (silicone) and degradable (polyanhydrides, poly-lactide-co-glycolide) biomatrices to deliver drugs for prolonged periods and have been successfully used to improve curcumin bioavailability [241].

Lately, delivery systems based on food macromolecules (whey protein, casein, food prolamine, gelatin nanoparticles) have been used to improve the bioavailability of resveratrol, curcumin, and EGCG [228]. Other strategies for improving polyphenol bioavailability include spray-drying in the case of apigenin or anthocyanins [243,246]. Alternative routes of administration (oral transmucosal administration or inhalable spray-dried formulation), together with novel formulations (hexose-resveratrol), have been used for the improvement of resveratrol bioavailability [244].

## 4. Conclusions

Polyphenols act as chemopreventive agents in prostate cancer due to their antioxidant or pro-oxidant effects (through reduction of ROS, increasing antioxidant enzyme activity, or inducing cytotoxicity), modulation of androgen receptors (inhibition of expression and function) or their transactivation of signaling pathways (PI3K, Akt, ERK1/2, FoxO, GSK-3β, RTK, etc). Furthermore, they induce cell cycle arrest (through downregulation of cyclin D1, cyclin E, CDK2, CDK4, and cdc25 and upregulation of proteins coded by tumor suppressor genes such as p53, p21, p27, etc.) and apoptosis (through activation of caspase-3, caspase-9, cytochrome *c*, or Bax proteins). Moreover, polyphenols have proven beneficial against tumor invasion through inhibition of angiogenesis and metastasis. The decline of metastatic tumors is due to a reduction in enzymes such as MMPs and uPA and an increase in E-cadherin, while the inhibition of angiogenesis is the consequence of reduced levels of VEGF. The main epigenetic mechanisms responsible for the antitumor effects of polyphenols involve a decrease in the hypermethylation of tumor suppressor genes and downregulation of oncogenic miRNA.

The role of polyphenols in both the treatment and prevention of prostate cancer depends on their bioavailability. Oral bioavailability was shown to be variable depending on the chemical structure of each compound, food matrix, interaction with other nutrients, and host-related factors. Interestingly, polyphenol metabolites, such as urolithins, equol, enterodiol, enterolactone, protocatechuic acid, 3-hydroxyphenyl propionic acid, 4-hydroxyphenylacetic acid, and hippuric acid play an important role against prostate cancer. However, most of the studies conclude that polyphenol bioavailability is relatively low. Therefore, the development of new strategies to enhance their bioavailability is a subject of present interest. A promising approach in this regard is the development of drug delivery systems that are able to release polyphenols in a controlled manner and in the target tissue. Other methods for increasing the bioavailability of polyphenols include solid dispersion techniques for microencapsulation, de novo formulations, or alternative routes of administration. Better knowledge about the bioavailability of polyphenols is essential to properly evaluate their role as chemopreventive agents in different types of cancer, and continuous research in this area is needed.

## Figures and Tables

**Figure 1 ijms-20-01062-f001:**
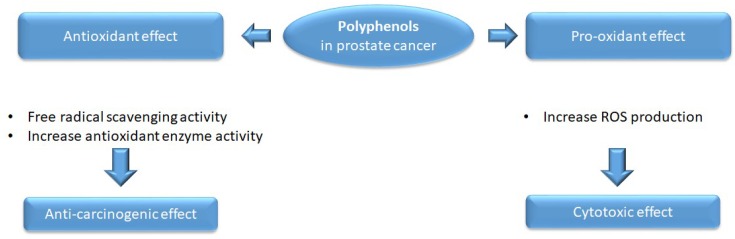
Antioxidant and pro-oxidant effects of polyphenols in prostate cancer. Polyphenols can act as both antioxidant molecules, by free radical scavenging, and as pro-oxidant agents, by increasing ROS production—the mechanisms are dependent on the concentrations applied. In addition, polyphenols are able to increase the level of antioxidant enzymes in prostate cancer cell lines or in animal models of prostate cancer.

**Figure 2 ijms-20-01062-f002:**
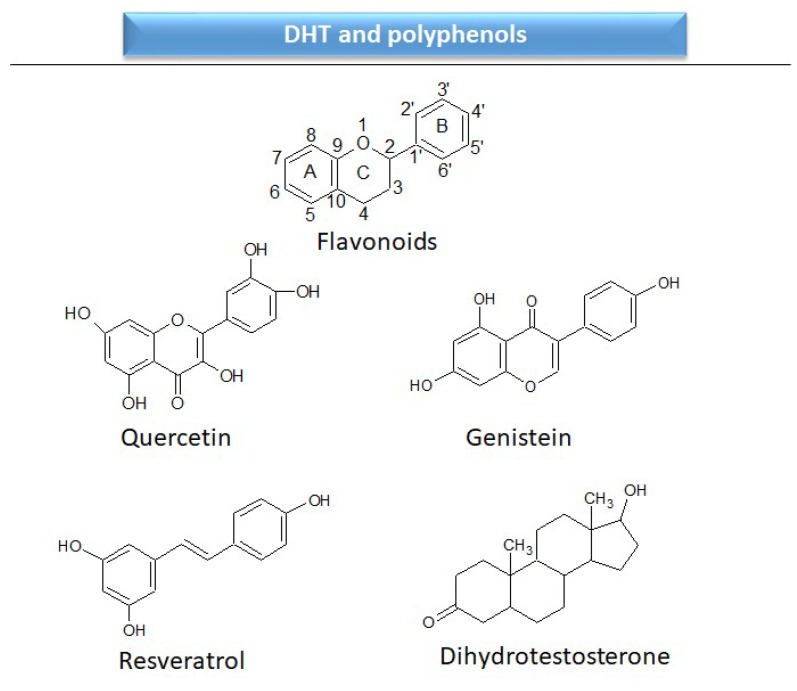
Chemical structures of some polyphenols that display similarities to dihydrotestosterone (DHT).

**Figure 3 ijms-20-01062-f003:**
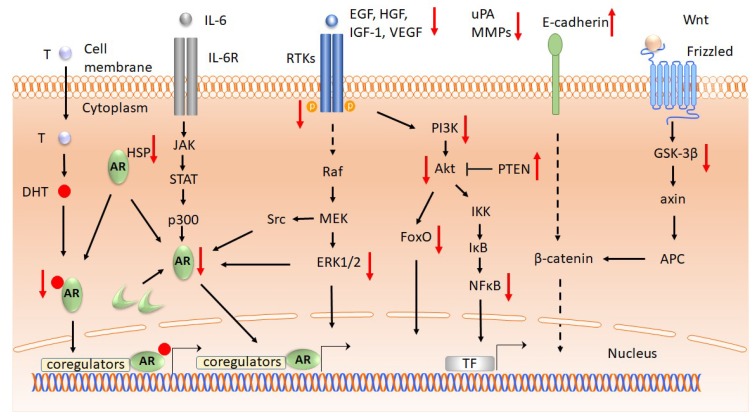
The effects of polyphenols on signaling pathways in prostate cancer. In the canonical pathway ARs are activated by DHT binding. AR can be transactivated in the absence of DHT through different signaling pathways: increased activity of RTK or interleukin receptors and their signaling pathways, modifications of the coregulatory proteins, or constitutively activated AR [115,126]. Ligand binding to RTK activates PI3K/Akt kinases, which trigger the activation of IKK. This pathway induces phosphorylation of IκB, resulting in its ubiquitylation and proteasome-mediated degradation. NF-κB is maintained in the cytoplasm by the interaction with IκB, and degradation of IκB activates NF-κB, which in turn is enabled to enter the nucleus and activate the genes involved in cell survival [127,128]. For simplification, the canonical pathway of NF-κB activation that takes place through the members of tumor necrosis factor receptors or interleukin receptors is omitted from the figure [129]. Similarly, the canonical pathway of PI3K/Akt signaling is not presented. Polyphenols might modulate the levels of signaling molecules in prostate cancer by decreasing or increasing their levels. Polyphenols downregulate AR (quercetin, genistein, resveratrol, EGCG), HSP90 (genistein), IGF-1 (apigenin), EGFR (curcumin, resveratrol), HER2 (resveratrol), ERK (apigenin, gallic acid, EGCG), phosphorylated PI3K (apigenin, curcumin, resveratrol), phosphorylated Akt (apigenin, CAPE, gallic acid, resveratrol), FoxO (apigenin), NF-κB (apigenin, curcumin, gallic acid, EGCG), GSK-3β (CAPE), VEFG (apigenin, genistein, quercetin, EGCG), uPA (apigenin), MMPs (apigenin, gallic acid, EGCG, genistein) and upregulate PTEN (resvetratol) and E-cadherin (apigenin). Legend: *AR*, androgen receptors; T, testosterone; DHT, dehydrotestosterone; HSP, heatshock protein; RTK, receptor tyrosine kinase; Raf, rapid accelerated fibrosarcoma protein; MEK, mitogen activated protein kinase, kinase; ERK1/2, extracellular signaling regulate d kinase; EGF, epidermal growth factor; IGF-1, insulin-like growth factor 1; HGF, hepatocyte growth factor; VEGF, vascular endothelial growth factor; PI3K, phosphoinositide 3 kinase; Akt, Ak thymoma protein-kinase (protein kinase B); NF-κB, nuclear factor kappa-light-chain-enhancer of activated B cells; IκB, inhibitor of κB; IKK, IκB kinase; FoxO, forkhead box O protein; PTEN, phosphatase and tensin homolog; Wnt, wingless/integrated ligand; GSK-3β, glycogen synthase kinase-3β; APC, adenomatous polyposis coli; TF, transcription factors; CAPE, caffeic acid phenethyl ester; EGCG, epigallocatechin gallate; upregulation (red ↑), downregulation (red ↓).

**Figure 4 ijms-20-01062-f004:**
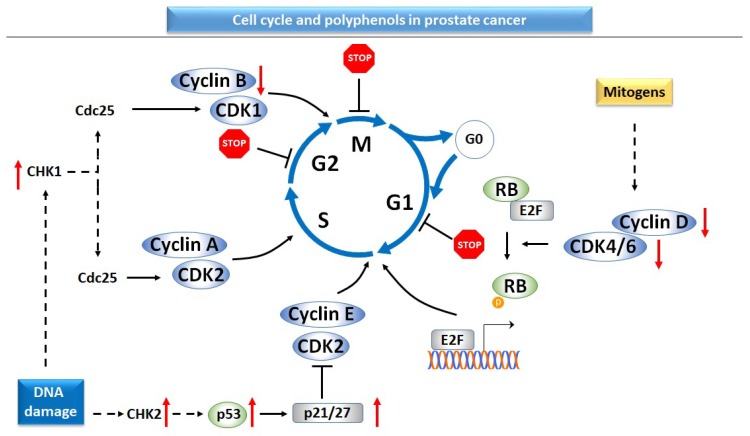
Cell cycle and the inhibitory effect of polyphenols in prostate cancer cells. DNA damage triggers the activation of the tumor suppressor and checkpoint proteins, while mitogen signals induce the progress of the cell through the cell cycle [136]. The following molecules are reported to be downregulated by polyphenols in prostate cancer: cyclin D (apigenin, caffeic acid phenethyl ester, gingerol), cyclin E (caffeic acid phenethyl ester, gingerol, quercetin), CDK2 (quercetin), CDK4 (gingerol), cdc25 (gallic acid). Polyphenols upregulate proteins coded by tumor suppressor genes, such as p53 (apigenin, curcumin, EGCG), p21 (apigenin, EGCG), p27 (apigenin) and checkpoint proteins CHK1, 2 (gallic acid). Moreover, administration of polyphenols induced the cell cycle arrest in a cell-line- and compound-dependent manner, particularly G2/M arrest (apigenin, gallic acid), G0/G1 arrest (EGCG, quercetin). Legend: cyclin-dependent kinases (CDK), retinoblastoma protein (RB), transcription factor E2 (E2F), mitosis (M), gap 1 phase (G1), gap 2 phase (G2), DNA synthesis phase (S), CHK, checkpoint proteins; cdc25, cell cycle division protein 25; upregulation (red ↑); downregulation (red ↓).

**Figure 5 ijms-20-01062-f005:**
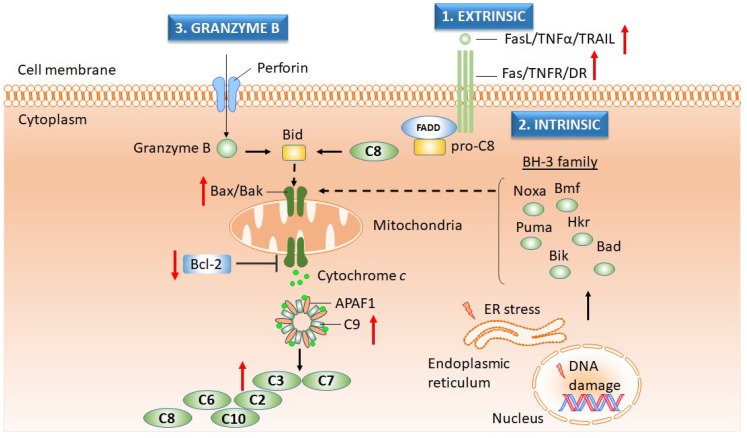
Apoptosis can be modulated by the activity of polyphenols in prostate cancer. Induction of apoptosis takes place through three main mechanisms: extrinsic, intrinsic, and perforin/granzyme pathways [145]. In the extrinsic pathway, activation of transmembrane death receptors (Fas/TNFR/DR) by their ligands (FasL/TNFα/TRAIL) induces the recruitment of FADD and further results in triggering of caspase-8 action. Mitochondrial release of cytochrome *c*, provoked by the truncated BID protein, induces the formation of the apoptosome and activation of caspase-9, which in turn activate caspase-3 and -7. The intrinsic pathway cross-talks with the extrinsic pathway at the mitochondrial level, where the BH-3 family of proteins (Noxa, Bmf, Puma, Hkr, Bik, etc.) promotes apoptosis through Bax/Bak assembly. The regulation of the apoptotic pathways is coordinated by the Bcl-2 family of proteins, which consists of both pro-apoptotic members (Bax, Bak) and anti-apoptotic members (Bcl-2). The cytotoxicity mediated by immune cells (i.e., cytotoxic T lymphocytes) occurs through pore formation in the membrane of the target cells. This pore can be formed with the help of pore-forming proteins (perforin) and is used to deliver proteases (granzymes), which in turn will trigger the apoptotic pathway [145,153]. The pro-apoptotic molecules that are upregulated by polyphenols in prostate cancer are caspase-3 (apigenin, gallic acid, gingerol, EGCG, quercetin), caspase-8 (apigenin, gallic acid, EGCG, quercetin), caspase-9 (apigenin, gallic acid, EGCG, quercetin), cytochrome c (gallic acid), Bax (apigenin, EGCG, quercetin), TRAIL (apigenin), and DR (apigenin), while the anti-apoptotic molecules that are downregulated are Bcl-2 (apigenin, gingerol, EGCG, and quercetin). Legend: FasL, Fas ligand; TNFα, tumor necrosis factor alpha; TRAIL, TNF-related apoptosis-inducing ligand; Fas, Fas receptor/ apoptosis antigen 1; TNFR, TNF receptor; DR, death receptor; FADD, Fas-associated death domain protein; Bcl-2, B-cell lymphoma type 2 protein; BH-3, Bcl-2 homology domain 3; Bid, BH3-interacting domain death agonist; Noxa, phorbol-12-myristate-13-acetate-induced protein 1; Bmf, Bcl-2 modifying factor; Puma, p53 upregulated modulator of apoptosis/ Bcl-2 binding component-3; Hkr, harakiri death protein; Bik, Bcl-2 interacting killer; Bad, Bcl-2 antagonist of cell death; Bax, Bcl-2-associated X protein; Bak, Bcl-2-antagonist/killer-1; ER, endoplasmic reticulum; upregulation (red ↑); downregulation (red ↓); schematic diagram was performed with Biomedical PPT Toolkit Suite, Motifolio, Inc., Ellicott Citty, MD, USA.

**Table 1 ijms-20-01062-t001:** Main classes of phenolic compounds with representative members and dietary sources frequently investigated in prostate cancer studies.

**Phenolic Compounds**	**Representative** **Compounds**	**Chemical Structure**	**Dietary Sources**	**Ref.**
**FLAVONOIDS**				
**Flavones**	apigenin	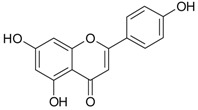	oranges, lemons, apricots, apples, black currants, bananas, potatoes, spinach, onions, lettuce, beans, cereals	[53,54]
**Flavonols**	quercetin	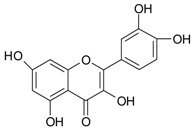		
**Flavan-3-ols**	epigallocatechin gallate (EGCG)	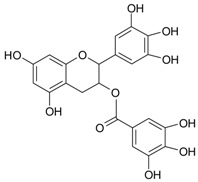	green/black tea	[53,54]
**Isoflavones**	genistein	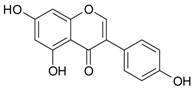	soy milk, tofu, *nattō*	[30,55]
**Anthocyanins**	cyanidin	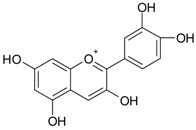	plums, grapes, elderberries, cherries	[55]
**Proantho-cyanidins**	proantho- cyanidin B	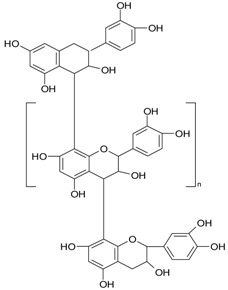	cranberries, grapes, walnuts, rice	[53]
**NON-FLAVONOID**				
**Hydroxy-benzoic acids**	gallic acid	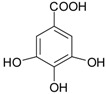	blackcurrants, strawberries, raspberries, kiwi, cherry, plums	[30,55,56,57]
**Hydroxy-cinnamic acids**	caffeic acid phenethyl ester	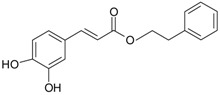	artichoke, oregano, thyme, basil, coffee, mushrooms, medicinal plants	[55]
**Ellagitannins** **(ellagic acid derivatives)**	punicalagin	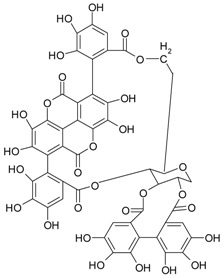	berry fruits pomegranate	[58,59]
**Stilbens**	resveratrol	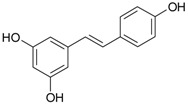	grapes, mulberries	[60]
**Lignans**	secoisolarici-resinol	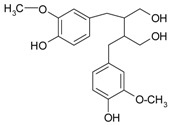	sesame, flaxseeds	[61]
**Other compounds**	curcumin	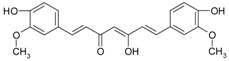	*Curcuma* roots	[62]
	gingerol	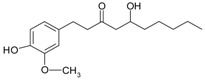	fresh/dried ginger rhizomes	[63]

**Table 2 ijms-20-01062-t002:** Main characteristics of cell lines used to study prostate cancer.

Cell Type	Characteristics	References
**Non-transformed prostate cell lines**		
PrEC	Normal prostate epithelial cells	[71,72]
RWPE-1	Non-malignant epithelial prostate cell line	[73]
**PCa cell lines**		
LNCaP	Androgen-responsive human prostate adenocarcinoma cell line; secrete PSA; low tumorigenicity in nude mice; have a mutated AR at T877A	[70,74,75]
PC-3	Androgen-independent human prostate adenocarcinoma cell line; obtained from bone metastasis of prostate adenocarcinoma	[70,76,77]
PC-3M	Metastatic androgen-independent human prostate adenocarcinoma cell line	[78]
PC-3M-MM2	Highly invasive androgen-independent human prostate adenocarcinoma cell line	[79]
DU-145	Androgen-independent human prostate adenocarcinoma cell line; metastatic cell line isolated from brain	[80]
22Rv1	Castration-resistant PCa cell line with hyper-diploid DNA (50 chromosomes); 22Rv1 cells express PSA	[81]
prostate CSC	CSC isolated from PC-3 cells positive for CD44^+^	[82]
CxR	Castration-resistant PCa cell line to cabazitaxel treatment (inhibitor of microtubule activity)	[83]
LAPC-4	Androgen-responsive human prostate adenocarcinoma cell line; established from lymph node metastasis in xenograft models from patients with advance disease	[84]
WPE1-NB14, WPE1-NB11, WPE-NA22	Cell lines of prostate adenocarcinoma; contain DNA from human papilloma virus 18	[85]
C4-2, C4-2B	Metastatic androgen-independent human prostate sublines derived from LNCaP cells; able to develop bone metastasis	[86]

**Table 3 ijms-20-01062-t003:** Molecular targets for the dietary polyphenols in prostate cancer.

Cellular Effect	Polyphenol	Molecular Target	Cell Line/ Animal Model/ Clinical Trial	References
**Antioxidant effect**	Quercetin	↓ROS ↑SOD, ↑CAT, ↑GPx, ↑GSR	Sprague‒Dawley rats	[100]
	Genistein	↑GPx	LNCaP, PC-3 cell lines	[99]
	EGGC	↑SOD	DU-145 cell line	[101,102]
**Pro-oxidant effect**	Apigenin	↑ROS	22Rv1 cell line	[150]
	Quercetin	↑ROS	DU-145 cell line	[97]
**Androgen and estrogen receptors**	Quercetin	↓AR	LNCaP cell line	[106,195,196]
	Genistein	↓AR (high doses of genistein) in correlation with ↓HSP90	Sprague‒Dawley rats LNCaP cell line	[66,107,197,198]
		↓ERα	‒-Dawley rats	[198]
		↑AR (physiological doses of genistein)	PC-3 cells transfected with T877A-AR	[107]
	Resveratrol	↓AR	LNCaP cell line HeLa cells transfected with human AR	[108,109,110,111,112]
		↓ERα	PC-3 cell line	[108]
	EGCG	↓AR, ↓ mRNA for AR	22Rv1 tumor xenograft in nude mice	[113]
**Growth factors and cytokines receptors**	Apigenin	↓IGF-1	TRAMP mice	[117]
	Resveratrol	↓EGFR, ↓HER2	LNCaP, C4-2 cell lines	[118]
	Curcumin	↓CXCL-1, -2 ↓EGFR (Tyr 845, Tyr 1068)	PC-3 cell line	[119,120]
	EGCG	↓c-Met/HGF (Tyr1234/1235)	DU-145 cell line	[121]
**Signal transduction**	Apigenin	↓PI3K, ↓p-Akt (Ser473, Thr308), ↓ERK1/2, ↓p-FoxO (Ser253), ↓NF-κB	22Rv1 TRAMP mice Prostate CSC (CD44^+^) isolated from PC-3 cells	[82,117,125,150]
	CAPE	↓ERK1/2, ↓p-Akt (Ser473), ↓p-mTOR (Ser2448, Ser24981), ↓p-GSK3α (Ser21), ↓p-GSK3β (Ser9), ↓p-PDK1 (Ser241)	LNCap, DU-145, PC-3 cell lines	[130,131]
	Curcumin	↓NF-κB ↓p-PI3K, ↓p-Akt (Ser 473, Thr 208), ↓p-IκB	LNCaP, PC-3 cell lines	[119,120,132]
	Gallic acid	↓SOS, ↓GRB2, ↓PKC, ↓NF-κB, ↓JNK, ↓ERK1/2, ↓p38-MAPK, ↓p-Akt	LNCaP, DU-145, PC-3 cell lines	[95]
	Gingerol	↓MRP1	PC-3 cell line	[134]
	EGCG	↓NF-κB, ↓ERK1/2, ↓p-Akt	DU-145 cell line	[90,121]
	Resveratrol	↓PI3K, ↓Akt, ↓GSK-3, ↑PTEN	LNCaP, PC-3 cell lines TRAMP mice	[108,118]
**Cell cycle**	Apigenin	↓cyclin D1 Arrest in G_0_/G_1_ or G_2_/M phase	LNCaP, PC-3 cell lines TRAMP mice	[125,139]
	CAPE	↓cyclin D1 ↓cyclin E	PC-3 cell line	[130]
	Gallic acid	G2/M arrest ↓cdc25 ↑CHK1, CHK2	LNCaP, DU-145, PC-3 cell lines	[95]
	Gingerol	↓cyclin D1, ↓cyclin E ↓CDK4	Normal prostate epithelial cells (PrEC) PCa cell lines: LNCaP, DU-145, PC-3, C4-2, C4-2B	[71]
	EGCG	G0/G1 arrest or G2/M arrest—cell-line-dependent	LNCaP, DU-145, PC-3 cell lines	[144]
	Quercetin	G0/G1 arrest ↓CDK2, ↓cyclin E ↓cyclin D	PC-3 cell lines	[68]
**Apoptosis**	Apigenin	↑caspase-3, -8 ↓*ΔΨ_m_* ↑cytochrome *c* ↓Bcl-2, ↓Bcl-XL ↑Bax ↑TRAIL, ↑DG5	22Rv1, PC-3 (p53^-/-^), PC-3(p53^+/+^) cell lines Prostate CSC (CD44^+^) isolated from PC-3 cells	[82,150,151]
	Gallic acid	↑cytochrome *c* ↑caspase-3, -8, -9	LNCaP cell line	[96]
	Gingerol	↑caspase-3, ↑PARP ↓Bcl-2	Normal prostate epithelial cells (PrEC) PCa cell lines: LNCaP, DU-145, PC-3, C4-2, C4-2B	[71]
	EGCG	↓Bcl-2, ↑Bax ↑caspase-3, -8, -9 ↑PARP ↑CHOP/GADD153	LNCaP, DU-145, PC-3 cell lines	[144,154]
	Quercetin	↓Bcl-2, ↑Bax ↑caspase-3, -8, -9 ↑ATF ↑GRP78 ↑GADD153	PC-3 cell line	[68,158]
**Invasion and metastasis**	Apigenin	↓uPA, ↓MMP-2, ↓MMP-9 ↑E-cadherin ↓vimentin	DU-145 cell line TRAMP mice	[117,139]
	EGCG, gallic acid, genistein	↓MMPs	PC-3, DU-145	[90,133,163,164,165]
**Angiogenesis**	apigenin, genistein, quercetin, EGCG	↓VEGF	PC-3 TRAMP mice Men of ages 18 to 75 years with PCa	[117,164,171,172]
**Oncogenes and the coding proteins**	Curcumin, EGCG	↓c-Jun (Ser73)	LNCaP, DU-145 cell lines	[90,177]
**Tumor suppressor genes and the coding proteins**	Apigenin	↑p53 ↑p27/Kip1 ↑p21/CIP1	22Rv1, LNCaP, PC-3 cell lines TRAMP mice	[125,150]
	Curcumin	↑p53	PC-3 cell line	[119]
	EGCG	↑p53, ↑p21/CIP1	LNCaP cell line	[154]
**DNA methylation and histone modification**	Curcumin	↓p300-HAT ↓H3 acetylation	PC-3M cell line	[184]
	Genistein	↓DNA methylation of RARβ ↓*BTG3* gene methylation	LNCaP, PC-3 cell lines	[185,186]
**miRNA**	EGCG	↓oncogenic miR-21 ↑tumor suppressor miR-330	LNCaP, 22Rv1 cell lines	[113]
	Genistein	↓oncogenic miR-151 ↑tumor suppressor miR-574-3p	LNCaP, PC-3, DU-145 PCa cell lines RWPE-1 non-malignant epithelial prostate cell line	[73]
	Resveratrol	↓oncogenic miR-21	Highly invasive PC-3M-MM2, DU-145, LNCaP cell lines	[79]

Legend: ROS, reactive oxygen species; SOD, superoxide dismutase; CAT, catalase; GPx, glutathione peroxidase; GSR, glutathione reductase; EGCG, epigallocatechin gallate; AR, androgen receptor; HSP90, heat shock protein 90; IGF-1, insulin-like growth factor 1; EGFR, epidermal growth factor receptor; HER2, receptor tyrosine kinase ErbB2/v-ErbB2 avian erithroblastic leukemia viral homolog 2; CXCL-1, -2, chemokine with CXC motif ligand -1, -2; c-Met/HGF, hepatocyte growth factor; PI3K, phosphatidylinositol 3-kinase; Akt, Ak tymoma protein/PKB, protein kinase B; ERK 1/2, extracelluar signal-regulated kinases -1, -2; FoxO, forkhead box O protein; NF-κB, nuclear factor kappa-light-chain-enhancer of activated B cells; mTOR, mammalian target of rapamacyn; GSK-3β, glycogen synthase kinase; PDK1, phosphoinositide-dependent kinase-1; IκBα, inhibitor of NF-κB; SOS, son of sevenless; GRB2, growth factor receptor-bound protein 2; PKC, protein kinase C, JNK, c-Jun N-terminal kinase; MAPK, mitogen activated protein kinase; MRP1, multidrug resistance-associated protein 2; PTEN, phosphatase and tensin homolog; cdc25, cell cycle division protein 25; CHK1, checkpoint kinase 1; caspase-3, cysteine-aspartic acid protease 3; ΔΨm, mitochondrial membrane potential; Bcl-2, B-cell lymphoma type 2 protein; Bcl-XL, Bcl-2 extralarge protein; Bax, Bcl-2-associated X protein; TRAIL, TNF-related apoptosis-inducing ligand; DG5, death receptor; PARP, poly(ADP-ribose) polymerase; CHOP, CCAAT-enhancer-binding protein homologous protein; GADD153, growth arrest and DNA damage inducible Protein 153 protein; ATF, activating transcription factor; GRP78, glucose regulated protein of 78 kDa; uPA, urokinase-type plasminogen activator; MMP-2, matrix metalloproteinase 2; VEGF, vascular endothelial factor; c-Jun, avian sarcoma virus 17 homolog; p27/Kip1, kinesin-like protein; p21/CIP1, cyclin-dependent kinase inhibitor 1A/CDK-interacting protein 1; RARβ, retinoic acid receptor beta; BTG3, B-cell translocation gene; miR, microRNA.

**Table 4 ijms-20-01062-t004:** Role of polyphenol metabolites in prostate cancer.

Polyphenols	Metabolite	Cell Lines/ Animal Model/ Clinical Trial	Cellular Mechanism	Molecular Target	Reference
Ellagitannins	UA, EA	DU-145, PC-3 cell lines	Cell cycle	G2/M phase ↑cyclin B1/ cdc2 phosphorylation (UA) S phase ↓cyclin B1, cyclin D1 (EA)	[209]
Ellagitannins	UAA, UAB	LNCaP, PC-3 cell lines	Apoptosis AR	↓Bcl-2 ↓AR expression ↓PSA	[46]
Ellagitannins and green tea catechins	UAA and M4	LNCaP cell line	AR	↓AR expression ↓PSA	[221]
Ellagitannins	EA	LNCaP cell line	Angiogenesis	↓OH1, OH2 ↓VEGF, ↓OPG	[222]
Lignans	EL, ET	Phase II randomized control trial in PCa men awaiting prostatectomy	Angiogenesis Inhibition of proliferation	↓VEGF, ↓Ki-67	[44]
	EL	RWPE-1, WPE1-NA22, WPE1-NB14, WPE1-NB11, WPE1-NB26, LNCaP cell lines	miRNA, tumor suppressor genes	↑PTEN, ↓miR-106b cluster	[223]
Isoflavones	equol	PC-3, DU-145, LNCaP, CxR, 22Rv1 cell lines	Signal transduction AR	↑Akt/FOXO3a, ↓AR through Skp2 pathway	[48,49]
	O-DA	LNCaP, LAPC-4 cell lines	AR	↓AR	[225]
Antho- cyanins	PrA	LNCaP cell lines	Apoptosis Angiogenesis Anti-inflamma-tory effects	↑caspase -3, ↓membrane mitochondrial potential, ↓VEGF, ↓IL-6, IL-8	[43]
Flavan-3-ols, proanthocya- nidins	Hipp 3-Hppp 4-Hpa	LNCaP cell line	Cell cycle	↓cyclin B1	[207]

Legend: PSA, prostate specific antigen; cyclin B1/cdc2 phosphorylation, mitosis-promoting factor; AR, androgen receptor; O-DA, O-desmethyl angolensin; UA, urolithins; UAA, urolithin A; UAB, urolithin B; M4 (3′, 4′, 5′–trihydroxyphenyl)–γ-valerolactone; EL, enterolactone; ED, enterodiol; OH1, OH2, heme oxygenase system; VEGF, vascular endothelial growth factor; OPG, osteoprotegerin; PTEN, phosphatase and tensin homolog; FOXO3a, forkhead box O protein; Skp2, S-phase kinase-associated protein 2; PrA, protocatechuic acid; IL-6, IL-8, interleukins 6, 8; Hipp, hippuric acid; 3-Hppp, 3-hydroxyphenyl propionic acid; 4-Hpa, 4-hydroxyphenylacetic acid.

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
