# Peer review of "Molecular Mechanisms and Bioavailability of Polyphenols in Prostate Cancer"

_ijms, 2019, doi:10.3390/ijms20051062_

Reviewer 1 Report

The review is generally very good.

A few recommendations-

I thinks a little more introduction is required to make the case for considering polyphenols a major item that needs to be explored.  Also, can you explain your criteria for so listed "polyphenols".

There are some lines that appear to be headings that are not separate in the text.

Eg. line 165-Generation of ----------.  Please check and either make into complete sentences or headings.

While the first 3/4th of the review is very good, toward the end there is greater reference to prostate cancer cells.  Please clarify if this means  cell lines or primary tumor cells.

Is it possible to add another column to Table 1 or add one more table to illustrate which prostate cell line or other materials were used demonstrate the points about the molecular targets?    

Author Response

Response to Reviewer 1 Comments

Manuscript ID: ijms-449674

Comments and Suggestions for Authors

General comment: The review is generally very good.

Response: We would like to thank you for all your comments and time commitment. We carried out the changes in the manuscript according to all your recommendations. The changes are made in the text, using Track Changes. Please, see below the answers point-by-point to your observations.

A few recommendations-

Point 1: I think a little more introduction is required to make the case for considering polyphenols a major item that needs to be explored. Also, can you explain your criteria for so listed "polyphenols".

Response 1: Introduction chapter was re-written and up-dated. The first three chapters are sum-up in a single introductory chapter. At the recommendation of second reviewer, the detailed figures about polyphenols are removed. We prepared a new table only with representative molecules frequently studied in prostate cancer experiments. The following paragraph about polyphenol classification is inserted in introduction:

”Polyphenols are classified as flavonoid and non-flavonoid compounds (Abdal Dayem, Choi et al. 2016, Amararathna, Johnston et al. 2016, Alam, Almoyad et al. 2018), according to the number of aromatic rings and the structural elements that bind these rings together (Ignat, Volf et al. 2011). Flavonoids have a C6-C3-C6 structural backbone and are subdivided according to their hydroxylation pattern and variations in the chroman ring into flavones, flavonols, flavanones and flavan-3-ols (Tsao 2010, Khoddami, Wilkes et al. 2013).”

Point 2: There are some lines that appear to be headings that are not separate in the text.

Eg. line 165-Generation of ----------.  Please check and either make into complete sentences or headings.

Response 2.1: The following statement ”Generation of ROS by the polyphenos through their metal chelator activity.” was removed.

Response 2.2: Second title ”invasion and metastasis” was replaced by ”angiogenesis”

Point 3: While the first 3/4th of the review is very good, toward the end there is greater reference to prostate cancer cells.  Please clarify if this means cell lines or primary tumor cells.

Is it possible to add another column to Table 1 or add one more table to illustrate which prostate cell line or other materials were used demonstrate the points about the molecular targets?    

Response 3: The data from subchapter “Role of polyphenol metabolites in prostate cancer” was up-date with a table, which include the cell lines/animal models/clinical trials presented.

Table 4: Role of polyphenols metabolites in prostate cancer

Polyphenols

Metabolite

Cell   lines/

Animal   model/

clinical   trial

Cellular

mechanism

Molecular   target

Reference

Ellagitannins

UA, EA

DU-145, PC-3 cell lines

Cell cycle

G2/M phase

↑cyclin B1/

cdc2

phosphorylation (UA)

S phase

↓cyclin B1, cyclin D1 (EA)

[306]

Ellagitannins

UAA, UAB

LNCaP, PC-3 cell lines

Apoptosis

AR

↓Bcl-2

↓AR expression

↓PSA

[312]

Ellagitannins and green tea catechins

UAA and

M4

LNCaP cell line

AR

↓AR expression

↓PSA

[313]

Ellagitannins

EA

LNCaP cell line

Angiogenesis

↓OH1, OH2

↓VEGF, ↓OPG

[314]

Lignans

EL, ET

Phase II randomized control trial in PCa   men awaiting prostatectomy

Angiogenesis

Inhibition of proliferation

↓VEGF, ↓Ki-67

[307]

EL

RWPE-1, WPE1-NA22, WPE1-NB14, WPE1-NB11,   WPE1-NB26, LNCaP cell lines

miRNA,

tumor suppressor genes

↑PTEN,

↓miR-106b cluster

[315]

Isoflavones

equol

PC-3, DU-145, LNCaP, CxR, 22Rv1 cell lines

Signal transduction

AR

↑Akt/FOXO3a,

↓AR through Skp2 pathway

[308,   309]

O-DA

LNCaP, LAPC-4 cell lines

AR

↓AR

[317]

Antho-

cyanins

PrA

LNCaP cell lines

Apoptosis

Angiogenesis

Anti-inflama-tory effects

↑caspase -3,

↓membrane mitochondrial potential,

↓VEGF,

↓IL-6, IL-8

[310]

Flavan-3-ols, proanthocya-nidins

Hipp

3-Hppp

4-Hpa

LNCaP cell line

Cell cycle

↓cyclin B1

[232]

Legend: PSA, prostate specific antigen; cyclin B1/cdc2 phosphorylation, mitosis promoting factor; AR, androgen receptor; O-DA, O-desmethyl angolensin; UA, urolithins; UAA, urolithin A; UAB, urolithin B; M4 (3’, 4’, 5’–trihydroxyphenyl)–γ-valerolactone; EL, enterolactone; ED, enterodiol; OH1, OH2, heme oxygenase system; VEGF, vascular endothelial growth factor; OPG, osteoprotegerin; PTEN, phosphatase and tensin homolog; FOXO3a, forkhead box O protein; Skp2, S-phase kinase-associated protein 2; PrA, protocatechuic acid; IL-6, IL-8, interleukins 6, 8; Hipp, hippuric acid; 3-Hppp, 3-hydroxyphenyl propionic acid; 4-Hpa, 4-hydroxyphenylacetic acid.

Table about “Molecular targets for the dietary polyphenols in prostate cancer” was up-dated with a new column about cell lines/animal models/clinical trials.

Additionally, in the beginning of the chapter about molecular mechanisms, we inserted a new table with a brief characterization of the cell lines presented in the paper: 

Table 2: Main characteristics of cell lines used to study prostate cancer

Cell type

Characteristics

References

Non-transformed prostate   cell lines

PrEC

Normal prostate epithelial cells

[86, 87]

RWPE-1

Non-malignant epithelial   prostate cell line

[88]

PCa cell lines

LNCaP

Androgen responsive human   prostate adenocarcinoma cell line; secrete PSA; low tumorigenicity in nude   mice; have a mutated AR at T877A

[85, 89, 90]

PC-3

Androgen independent human   prostate adenocarcinoma cell line; obtained from bone metastasis of prostate   adenocarcinoma

[85, 91, 92]

PC-3M

Metastatic androgen   independent human prostate adenocarcinoma cell line

[93]

PC-3M-MM2

Highly invasive androgen   independent human prostate adenocarcinoma cell line

[94]

DU-145

Androgen independent human   prostate adenocarcinoma cell line; metastatic cell line isolated from brain

[95]

22Rv1

Castration resistant PCa   cell line with hyper-diploid DNA (50 chromosomes); 22Rv1 cells express PSA

[96]

prostate CSC

CSC isolated from PC-3   cells positive for CD44+

[97]

CxR

Castration resistant PCa   cell line to cabazitaxel treatment (inhibitor of microtubule activity)

[98]

LAPC-4

Androgen responsive human   prostate adenocarcinoma cell line; established from lymph node metastasis in   xenograft models from patients with advance disease

[99]

WPE1-NB14, WPE1-NB11, WPE-NA22

Cell lines of prostate   adenocarcinoma; contain human papilloma virus 18 DNA

[100]

C4-2, C4-2B

Metastatic androgen   independent human prostate sublines derived from LNCaP cells; able to develop   bone metastasis

[101]

Observation about bioavailability of polyphenols in prostate cancer: The chapter about bioavailability suffered major changes due to recommendations coming from the second reviewer, namely “although interesting, the section “5.1. Host related factors involved in polyphenols bioavailability” is apparently incoherent with the first part. I advise the authors to focus their attention on the effects of polyphenols in prostate cancer cells.”

Under these circumstances the chapter about bioavailability of polyphenols in prostate cancer include now only “role of polyphenol metabolites in prostate cancer” and “strategies for enhancing polyphenol bioavailability”. Also, we included the following statements in the beginning of the bioavailability chapter “Excellent reviews describe the bioavailability of polyphenols in humans [2004, Manach et al., Am J Clin Nutr; 2005, Manach et al., Am J Clin Nutr] and this is beyond the aim of present paper. Further, we will focus on the effect of polyphenols in prostate cancer, particularly on the role of polyphenols metabolites in prostate cancer together with strategies for enhancing polyphenols bioavailability.”

Thank you again for your feedback and time. 

Sincerely, 

Magda Mocanu, PhD

Reviewer 2 Report

In the present manuscript, authors summarized the molecular mechanisms of polyphenols actions in prostate cancer, androgen receptors (AR), key molecules involved in AR signaling and their transactivation pathways in cell cycle, apoptosis, angiogenesis and etc.

The reading of the manuscript in the present form is very hard. Authors reported a lot of information, frequently redundant. Lines 88-89 show a run on sentence. The aim is reported in lines 50-52.

Lines 207-209 show a run on sentence.

Introduction section doesn’t introduce correctly the topic of the review. In the introduction section, authors could deepen the charateristics of prostate cancer (PC) about the role of cancer stem cells and steroid receptors, as AR andER (I suggest the reading of recently published reviews- doi: 10.3389/fonc.2018.00002; doi: 10.18632/oncotarget.6220 to do a more comprehensive description of PC) and show the relation between PC and polyphenols. In my opinion, paragraphs 1, 2 and 3 should be collected in one paragraph.

The detailed description of the flavonoids and non-floavonoids sctructure and their sources is not pertinent.  I suggest to remove the Fig.1 and Fig. 2 and indicate only a representative structure of polyphenols. Authors should focus thier attention on the mechanisms of action and functions of polyphenols.

Althought interesting, the section on “5.1. Host related factors involved in polyphenols bioavailability” is apparently inchoerent with the first part. I advise the authors to focus their attention on the effects of polyphenols in prostate cancer cells.

Extended english revision is needed.

Author Response

Response to Reviewer 2 Comments

Manuscript ID: ijms-449674

Comments and Suggestions for Authors

General comment: In the present manuscript, authors summarized the molecular mechanisms of polyphenols actions in prostate cancer, androgen receptors (AR), key molecules involved in AR signaling and their transactivation pathways in cell cycle, apoptosis, angiogenesis and etc.

Response: We would like to thank you for all your comments and time commitment. We carried out the changes in the manuscript according to all your recommendations. The changes are made in the text, using Track Changes. Please, see below the answers point-by-point to your observations.

Point 1: The reading of the manuscript in the present form is very hard. Authors reported a lot of information, frequently redundant. Lines 88-89 show a run on sentence. The aim is reported in lines 50-52. Lines 207-209 show a run on sentence.

Response 1: To simplify the content of the manuscript and to make it easy to follow, we focused only on molecular mechanisms and bioavailability of the polyphenols in prostate cancer. In this context, the general information about polyphenol bioavailability was removed. Please see the up-date version of the manuscript.

Run on sentence:

“In view of epidemiology studies that revealed an inverse correlation between chronic diseases, such as cardiovascular diseases, cancer, diabetes mellitus or neurodegenerative diseases and diet rich in polyphenols, our paper will highlight the molecular mechanisms and bioavailability of the polyphenols in prostate cancer”

was replaced by

“Several epidemiology studies revealed an inverse correlation between chronic diseases and diet rich in polyphenols”.

The statement:

We propose here an equilibrated life style with a diet rich in dietary polyphenols as prophylactic attempts to slow down progression of localized prostate cancer or to prevent the occurrence of the disease.”

was removed from the introduction and was inserted in the abstract:

“Prostate cancer is the one of the most frequently diagnosed cancer among men over the age of 50. Several lines of evidence support the observation that polyphenols have preventive and therapeutic effects in prostate cancer. Moreover, prostate cancer is ideal for chemoprevention, due to its long latency. We propose here an equilibrated life style with a diet rich in polyphenols as prophylactic attempts to slow down progression of localized prostate cancer or to prevent the occurrence of the disease.”

Run on sentence from “AR” subchapter:

“Nevertheless, some of the prostate cancers, become “androgen refractory” during ADT, probably due to mutations or amplification of AR gene, which lead to AR overexpression and overactivation”

was replaced by:

“Nevertheless, some of the prostate cancers, become “androgen refractory” during ADT, probably due to mutations or amplification of AR gene”

Point 2: Introduction section doesn’t introduce correctly the topic of the review. In the introduction section, authors could deepen the characteristics of prostate cancer (PC) about the role of cancer stem cells and steroid receptors, as AR andER (I suggest the reading of recently published reviews- doi: 10.3389/fonc.2018.00002; doi: 10.18632/oncotarget.6220 to do a more comprehensive description of PC) and show the relation between PC and polyphenols. In my opinion, paragraphs 1, 2 and 3 should be collected in one paragraph.

Response 2: Thank you very much for the recent papers you recommended to us. Introduction chapter was re-written and up-dated. As you recommended, the first three chapters are sum-up in a single introductory chapter. Please, see the up-dated version of the manuscript.

Point 3: The detailed description of the flavonoids and non-flavonoids structure and their sources is not pertinent.  I suggest to remove the Fig. 1 and Fig. 2 and indicate only a representative structure of polyphenols. Authors should focus their attention on the mechanisms of action and functions of polyphenols.

Response 3: The detailed figures about polyphenols are removed from the manuscript. We prepared a short table only with representative polyphenols frequently studied in prostate cancer experiments.

Point 4: Although interesting, the section on “5.1. Host related factors involved in polyphenols bioavailability” is apparently incoherent with the first part. I advise the authors to focus their attention on the effects of polyphenols in prostate cancer cells.

Response 4: The chapter about bioavailability of polyphenols in prostate cancer is modified according to your recommendations. The detailed explanations about general bioavailability of polyphenol is removed from the manuscript. This chapter include now only “role of polyphenol metabolites in prostate cancer” and “strategies for enhancing polyphenol bioavailability”. To clarify this, we included the following statements in the beginning of bioavailability chapter “Excellent reviews describe the bioavailability of polyphenols in humans [2004, Manach et al., Am J Clin Nutr; 2005, Manach et al., Am J Clin Nutr] and this is beyond the aim of present paper. Further, we will focus on the effect of polyphenols in prostate cancer, particularly on the role of polyphenols metabolites in prostate cancer together with strategies for enhancing polyphenols bioavailability.”

Thank you again for your feedback and time. 

Sincerely, 

Magda Mocanu, PhD

Round  2

Reviewer 2 Report

The authors improved the quality of the manuscript revising extenstively the text, according to my suggestions. I accept the manuscript in the present form. I congratulate the authors for the efforts done.